# Discovery of bilaterian-type through-guts in cloudinomorphs from the terminal Ediacaran Period

James D. Schiffbauer [1,2]*, Tara Selly [1,2]*, Sarah M. Jacquet [1], Rachel A. Merz[3], Lyle L. Nelson[4], Michael A. Strange[5], Yaoping Cai[6] & Emily F. Smith [4]

The fossil record of the terminal Ediacaran Period is typified by the iconic index fossil *Cloudina* and its relatives. These tube-dwellers are presumed to be primitive metazoans, but resolving their phylogenetic identity has remained a point of contention. The root of the problem is a lack of diagnostic features; that is, phylogenetic interpretations have largely centered on the only available source of information—their external tubes. Here, using tomographic analyses of fossils from the Wood Canyon Formation (Nevada, USA), we report evidence of recognizable soft tissues within their external tubes. Although alternative interpretations are plausible, these internal cylindrical structures may be most appropriately interpreted as digestive tracts, which would be, to date, the earliest-known occurrence of such features in the fossil record. If this interpretation is correct, their nature as one-way through-guts not only provides evidence for establishing these fossils as definitive bilaterians but also has implications for the long-debated phylogenetic position of the broader cloudinomorphs.

[1] Department of Geological Sciences, University of Missouri, Columbia, MO 65211, USA. [2] X-ray Microanalysis Core, University of Missouri, Columbia, MO 65211, USA. [3] Biology Department, Swarthmore College, Swarthmore, PA 19081, USA. [4] Department of Earth and Planetary Sciences, Johns Hopkins University, Baltimore, MD 21218, USA. [5] Department of Geoscience, University of Nevada, Las Vegas, Las Vegas, NV 89154, USA. [6] Shaanxi Key Laboratory of Early Life and Environment, State Key Laboratory of Continental Dynamics, and Department of Geology, Northwest University, Xi'an 710069, China.
*email: schiffbauerj@missouri.edu; sellyt@missouri.edu

Commonly envisaged as a prelude to the Cambrian Explosion, the terminal interval of the Ediacaran Period (~550–539[1] million years ago (Ma)) chronicles several monumental events during the evolutionary dawn of animal life. Among the most significant are the emergences of biomineralization[2] and active motility[3], which demarcate this interval from the rest of the Ediacaran Period. Toward the Period's conclusion, the first metazoan mass extinction event[4,5] encompassed the downfall of the archetypal Ediacaran biota. Their demise, however, was coincident with an ecological shift in which organisms such as *Cloudina* and other occupants of this novel tube-building morphotype[4] become increasingly populous. Collectively, these "cloudinomorphs" (to avoid conflating unresolved phylogenetic relationships with shared morphologies[6]) were small, sessile, and epibenthic, but they appeared with several key adaptations that may have enhanced their chances for ecological success. These attributes include: (i) the advent of macroscopic biomineralization in the form of shelly external tubes[2], potentially serving as an impediment to predation[7]; (ii) the establishment of gregarious habits that may signal the onset of metazoan ecosystem engineering behaviors[8] (but see also ref. [9]); and (iii) the development of enhanced larval dispersal mechanisms and presumably both sexual and asexual reproductive habits[10] versus stolon-like reproductive modes of some members of the enigmatic soft-bodied "Ediacara biota"[11]. These compounded ecological innovations may have helped to place the cloudinomorphs as central players in ushering in a phase of fundamental ecosystem reform and increased trophic complexity[12]. Although its cause is equivocal at present, the changing of the ecological guard from largely sedentary Ediacara-type communities to much more dynamic syn-"Cambrian Explosion" ecosystems was well underway in the terminal Ediacaran. Indeed, as recently proposed[4], this interval possibly displays an even larger step-change in organismal and ecological complexity than at the Ediacaran–Cambrian boundary itself. Nonetheless, the most crucial task that remains is to untangle the potential relationships between the organisms of the Ediacaran Period and those well-defined as metazoans in the Cambrian Period. The cloudinomorphs are one of the few groups known to span the Ediacaran–Cambrian boundary[13], and thus understanding their phylogenetic position is key to unraveling the evolutionary and ecological relationships between the seemingly disparate biomes of the Ediacaran and Cambrian Periods.

The phylogenetic position of the cloudinomorphs has yet remained unresolved, albeit not without effort. Previous attempts[2,14–21] have used the only available information (to date) from the fossil record—their external tubes. Such features, specifically demonstrated by *Cloudina*, that have been employed to help constrain their phylogeny include (but are not limited to): (i) nested funnel-in-funnel tube construction[14]; (ii) smooth inner tube wall lumen[14]; (iii) presence of daughter-tube branching[10,14]; (iv) ovate tube cross-sections[15]; (v) bulbous shape of the closed posterior bases[10,14,16,17]; (vi) absence of basal attachment structures[15]; (vii) calcareous composition in mineralized representatives[14,19,22]; and (viii) microgranular tube wall ultrastructures[14]. There are several caveats that should be considered, however. First and foremost, some of these features are not uniformly representative across all of the cloudinomorphs—which should serve as a caution toward future attempts to resolve relationships within this morphotypic group. Moreover, at least some of these alleged diagnostic features (or lack thereof) may be taphonomic noise rather than primary biological signal. For instance, although a homogenous microgranular tube ultrastructure is commonly reported for *Cloudina*[14–16], lamellar construction has also been observed[16,23], raising the question as to the influence of diagenetic recrystallization on retention of primary ultrastructure[24], or, for that matter, original

composition[23]. It thus follows that the degree of tube wall biomineralization in addition to the original biomineral chemistry has been met with differing interpretations[19,22,23], likely compounded by varying preservational and diagenetic histories between localities[23,24]. The absence of substrate attachment structures[15] may be a consequence of displacement and transport during storm events[24,25], a common mode of deposition of cloudinomorphs that yields fragmental tubes in detrital hash resembling biohermal or reefal buildups[9]. Alternatively, if the attachment structures were originally soft tissue, they may have been taphonomically lost, in which case the absence of evidence should not be construed as evidence for absence. Ovate cross-sections may result from compression of a modestly flexible tube during sediment compaction, which is almost certainly the case for tubes of some cloudinomorph taxa that are interpreted to have been originally organic[6,26,27]. As such, and as should be the case with all enigmatic fossils, attempts at phylogenetic assessment would be best suited to focus on taphonomically robust features or those that can be best determined to be biologically and taxonomically informative.

In conjunction with the influences of a complex and wide array of taphonomic histories, placement of the cloudinomorphs is further confounded when we consider the diversity of modern tube-building organisms and their assumed convergence of tube-dwelling habits[28]. Most agree that the cloudinomorphs are at least of "lower" (phylogenetically earlier branching) diploblastic metazoan-grade organization[17]. However, differences in the value with which the aforementioned characteristics are weighted in comparison with polyphyletic modern tube-builders can yield a broad assortment of plausible affinities—ranging from chlorophytes to triploblastic metazoans. Although more antiquated interpretations have included presumably poriferan-grade archaeocyathids[2,18], recent discussion urged not to discount a macroalgal affinity, owing to comparable annulated tubular morphologies observed in modern calcareous dasyclad algae[29]. Extinct microconchid lophophorates have also been offered as a possible analog on the basis of tube structure and shape[17]. Similarly, some pterobranch hemichordates produce dichotomous organic-walled tubes[14] with reasonably comparable morphologies, and thus may also warrant consideration. Other authors have instead refused to wedge the cloudinomorphs into any extant or extinct group—proposing otherwise that they occupy their own *incertae sedis* stem-metazoan family, Cloudinidae[20,21]. Satisfying the perceived majority of their exterior tube characteristics, however, most researchers currently fall into either anthozoan cnidarian[15,19] or polychaete annelid[2,14,16,18] camps, but further distinction has been hindered by the absence of preserved soft tissues.

Here, from fossils of the Wood Canyon Formation, Nye County, Nevada, USA (Fig. 1), we provide a detailed report of internal soft-tissue preservation within cloudinomorph fossils, and, moreover, one of the earliest reports of preserved internal anatomical structures in the fossil record. On the basis of the morphology and interpreted physiology of this soft-tissue structure, we suggest that this feature holds significant potential to shed new light on the phylogenetic placement of the cloudinomorphs.

## Results

**Wood Canyon cloudinomorphs**. The Wood Canyon fossil assemblage is dominated by cloudinomorphic forms (Fig. 2). These fossils, as well as others from nearby units, have been taxonomically compared[30] with the well-studied tubular fauna of the Gaojiashan Lagerstätte, South China[26] and, more recently, to lesser-known cloudinomorphs from the East European

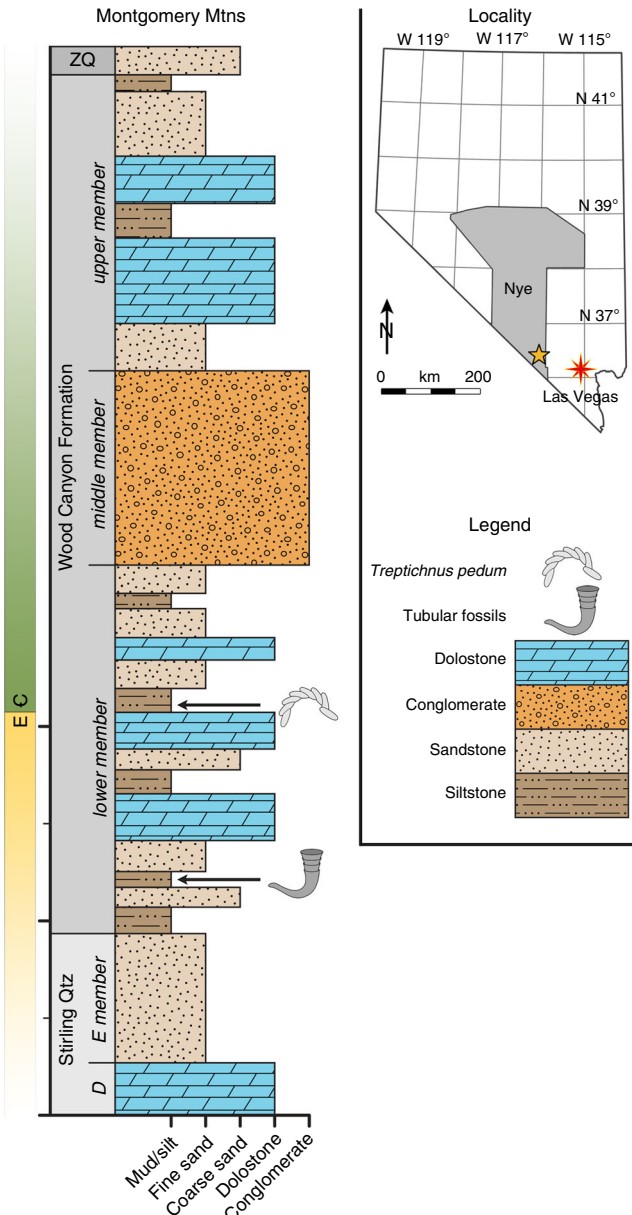

**Fig. 1 Generalized stratigraphy of the Montgomery Mountains site.** Ediacaran–Cambrian boundary denoted by the presence of *Treptichnus pedum*. Cloudinomorphs recovered from silty-shale below first dolostone marker bed in the lower member of the Wood Canyon Formation. Nye County indicated on map, with yellow star marking approximate sample locality. Stratigraphy after refs. [6,30]. ZQ = Zabriskie Quartzite; Stirling Qtz = Stirling Quartzite.

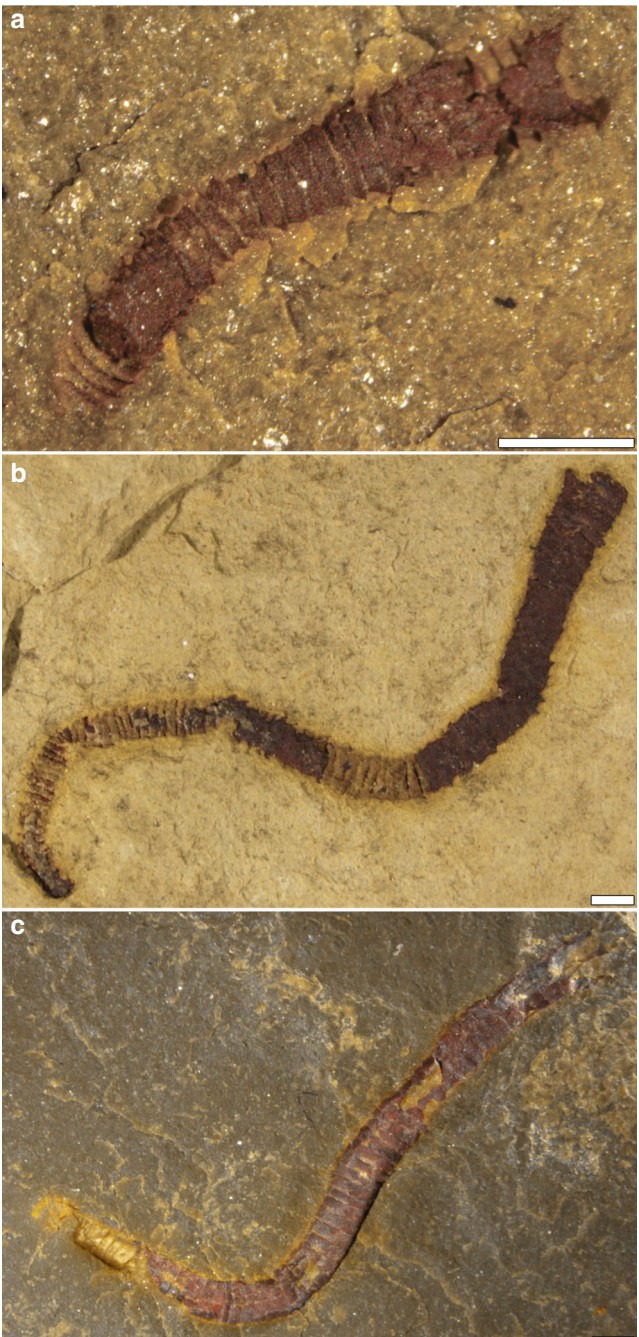

**Fig. 2 Wood Canyon cloudinomorphs of the Montgomery Mountains site.** **a** Holotype of *Saarina hagadorni*, sample USNM-E1636_009_B13. **b** Paratype of *S. hagadorni*, sample USNM-WCF_005_01. **c** Holotype of *Costatubus bibendi*, sample USNM-MS_DS_12. Samples reposited at the Smithsonian Institution. All scales = 1 mm, reproduced with permission from Selly, T. et al.[6] (in press) A new cloudinid fossil assemblage from the terminal Ediacaran of Nevada, USA. *Journal of Systematic Palaeontology*, https://doi.org/10.1080/14772019.2019.1623333.

Platform[6,27]. Systematic investigation of the Wood Canyon cloudinomorph fossils has thus far formally described two new species, *Saarina hagadorni* and *Costatubus bibendi*, as the most abundant in this locality[6]. Taphonomically, the Wood Canyon and Gaojiashan assemblages are highly comparable, with fossils from both units predominantly exhibiting three-dimensional pyritization[31]. However, whereas the majority of cloudinomorph tubes from the Gaojiashan are completely pyritized (e.g., the full tube volume is filled by pyrite mineralization)[31], those from Nevada show pyritized external tube walls retaining three-dimensionality but without pervasive pyrite infilling. As a result, the Nevadan cloudinomorphs offer a unique potential for

capturing resolvable soft tissues, and x-ray tomographic microscopy (μCT) provides an ideal method for non-invasive exploration of internal fossil features.

Unlike some of the cloudinomorphs that built more robust shelly tubes[14,17], the exterior tubes of the described Wood Canyon cloudinomorphs are inferred to have been organic in original composition from indications of plastic deformation[6],

much like the Gaojiashan taxon *Conotubus*[26] and East European representatives of *Saarina*[27]. Generic and specific taxonomic identification of the Nevadan tubular fossils containing soft tissues is unfortunately muddied by a lack of substantive exterior tube detail, likely resulting from chemical limitation during preservation (see "Preservational model" below). The soft-tissue-bearing tubular fossils exhibit exterior tube diameters (~2–4 mm) that generally fall within the observed range for the two described Wood Canyon cloudinomorph genera (maximum diameters = 3.92 mm and 6.36 mm for *Saarina hagadorni* and *Costatubus bibendi*, respectively), albeit greater than the median diameter for either genus (median diameters = 0.74 mm and 1.09 mm for *Saarina hagadorni* and *Costatubus bibendi*, respectively)[6]. We interpret the annulation of the tubes observed both optically and by µCT as a vestige of a "funnel-in-funnel" tube construction (Fig. 3), which supports the hypothesis of their cloudinomorphic affinities.

From three-dimensional reconstructions of µCT data, internal structures were revealed within the external tubes from a small subset of the analyzed specimens (~11%; 4 of 35 analyzed specimens; Figs. 3, 4, Supplementary Movies 1–3), which we here interpret as preserved soft tissues. The soft-tissue feature manifests as a sub-millimetric to millimetric diameter, centrally positioned cylinder that largely follows the curvature of the sagittal external tube length (Figs. 3, 4). In three of four cases, the cylindrical feature is mostly continuous through nearly the full length of the external tube (e.g., Fig. 3a), and only fragmented taphonomically (Fig. 3b). One of these specimens (Fig. 3c) shows significant kinking and sinuous bending of the internal cylinder relative to its external tube. The other specimen shows instead an incomplete internal cylinder (Fig. 4b), broken at a fragmented section of the external tube and also assumed to be unpreserved at the apical/posterior end of the external tube. When viewing the µCT data transversely to the tube length, the internal cylinder rests adjacent to the lower (with respect to bedding) internal surface of the tube wall (Fig. 4f).

To better explore the transverse morphology and preservation of the internal cylinders, a portion of the fragmented specimen (Fig. 4e) was selected for destructive preparation (via manual serial grinding) and subsequent scanning electron microscopic analyses. The sectioned soft-tissue cylinder was observed to be either infilled by sediment or fully mineralized (Fig. 5), and verified to be pyritic in composition. The external tube was additionally confirmed to have been pyritized (mostly weathered to iron oxyhydroxides), within a fine-grained siliciclastic host rock matrix (Fig. 5a). In cross-sectional view, the external tube can be complete (Fig. 5b), but appears more robustly pyritized at the bottom edge (see transverse slices in Supplementary Movie 2), and tenuous at the upper edge (Fig. 5c–e), with respect to bedding. Where the interior tube directly abuts the exterior tube, the exterior tube may be very thin (as observed in Fig. 5c, d), but this appears to be a localized phenomenon and is not apparent in all of the µCT- or SEM-observed (Fig. 5e) transverse cross-sections. The exterior tube shows marginal lateral compression (Fig. 5). In portions where the internal cylinder is broadly sediment-filled, it displays ovate cross-sections comparable in shape to the compressed external tube (Fig. 5a–c). In some of these sediment-filled portions, pyrite does exist within the interior of the tube, potentially replicating an organic template. Portions of the internal cylinder that are fully mineralized, in contrast, show circular, uncompressed cross-sections (Fig. 5d, e). Where the internal cylinder is sediment-filled, pyrite mineralization appears to extend both inward (towards the cylinder interior) and outward (into the lumen of the external tube) from a discernable cylinder wall (Fig. 6).

**Preservational model**. Each case of soft-tissue preservation presents a balance between taphonomically constructive and destructive processes, wherein retention and replication of biological information necessitates that decay does not eradicate, and mineralization does not overwrite, informative features. Impeding both decay and mineralization early in the taphonomic sequence of the Nevadan cloudinomorphs created a "goldilocks" scenario in which soft tissues may be distinguishably preserved, as opposed to their Gaojiashan contemporaries. Pyritization proceeds because of a confluence of chemical and microbiological factors, including: (i) a limited source of organic material (usually the soft tissues of the deceased organism); (ii) focused degradation of that organic material by sulfate-reducing bacteria; and (iii) anoxic pore waters rich in reduced iron along with available sulfate. While oxidizing the remnant organic material of the organism, sulfate-reducing bacteria (in normal seawater pH) convert sulfate to bisulfide, which then serves as one of the building blocks of pyrite along with reduced iron as the other[31].

If any part of this process becomes chemically starved, fossil pyritization will be halted. There are three paths that this can take, based on limitation of either organic matter, reduced iron, or sulfate. If bacterial sulfate reduction proceeds uninhibited by sulfate availability, the organics of the decaying organism are likely to be entirely consumed. This process, limited only by the availability of organics, would leave no soft tissues to be preserved, and should result in authigenic, centripetal pyrite infilling[31]. In the other two cases, pyritization can cease relatively early in the taphonomic sequence once the burial environment becomes chemically limiting (assuming no replenishment). If the availability of reduced iron is limited, pyrite formation will discontinue, but further degradation of the organics by sulfate reducers could continue unrestricted. Where sulfate concentration is instead limited, decay by sulfate-reducing bacteria would cease once the sulfate supply is expended. In turn, with no further generation of bisulfide, pyrite formation would be subsequently suspended once the available bisulfide is exhausted. Regardless which pathway is realized in the Wood Canyon burial environment, the necessary ingredient to preserve these soft tissues, and have them remain perceivable, is to terminate pyritization before overgrowth can obscure or homogenize the features.

In the Gaojiashan, pyritization likely proceeded uninhibited by sulfate or reduced iron[31–33]. Thus, even though the external tube morphology may be faithfully replicated in this assemblage, any internal structures were homogenized or obliterated by the combination of continued decay and mineralization. Conversely, we infer that pyritization of the Nevadan cloudinomorphs was abbreviated early in the taphonomic sequence by sulfate or reduced iron limitation. To briefly summarize taphonomy in the Wood Canyon (see also Fig. 7): (i) The initial burial event emplaced the cloudinomorphs within the sulfate reduction zone of the sediment (oriented prone to bedding, whether[34] or not[35] this was their in-vivo position). (ii) Decay by sulfate-reducing bacteria commenced, producing bisulfide that initiated pyrite mineralization. (iii) In a significantly sulfate-restricted local environment (with no sulfate replenishment), we infer that the rate of bacterial sulfate reduction may have also been diminished once sulfate concentrations dropped below rate-independent levels[36]. With tempered bacterially mediated decay, the earliest stages of mineralization focused on the two most histologically suitable loci for pyrite nucleation—the robust organic walls of the exterior tube and the presumably more labile internal soft-tissue cylinder. We suggest that pyrite mineralization of the external tube and internal cylinder occurred nearly simultaneously, as evidenced by the observed similarity in their compressed, ovate cross-sections from sediment compaction. (iv) Once structural

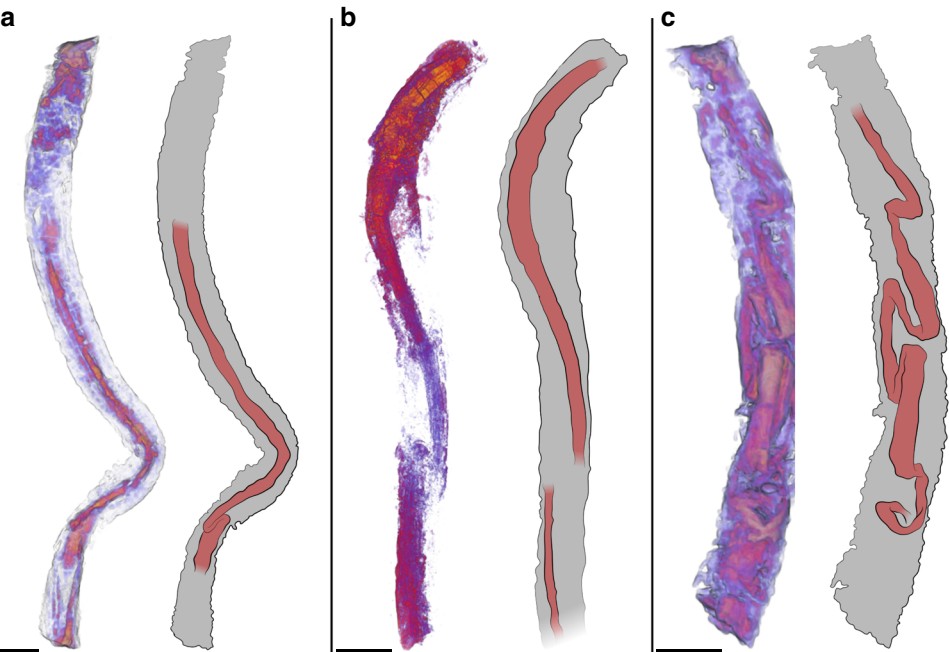

**Fig. 3 Soft tissue-bearing cloudinomorphs with schematic interpretation.** 3D volume render from µCT data shown in left image per frame (red-to-orange coloration indicates high density internal regions within exterior tube), with interpretive diagram in right image per frame. Examples here show **a** medial position and consistency (sample USNM-N1601_FL_018), **b** partial degradation/fragmentation (sample USNM-E1630_006), and **c** kinking and folding (sample USNM-N1601_FL_017). Soft tissue in sketches highlighted in red. Samples reposited at the Smithsonian Institution. All scales = 2 mm, sketches provided by Stacy Turpin Cheavens.

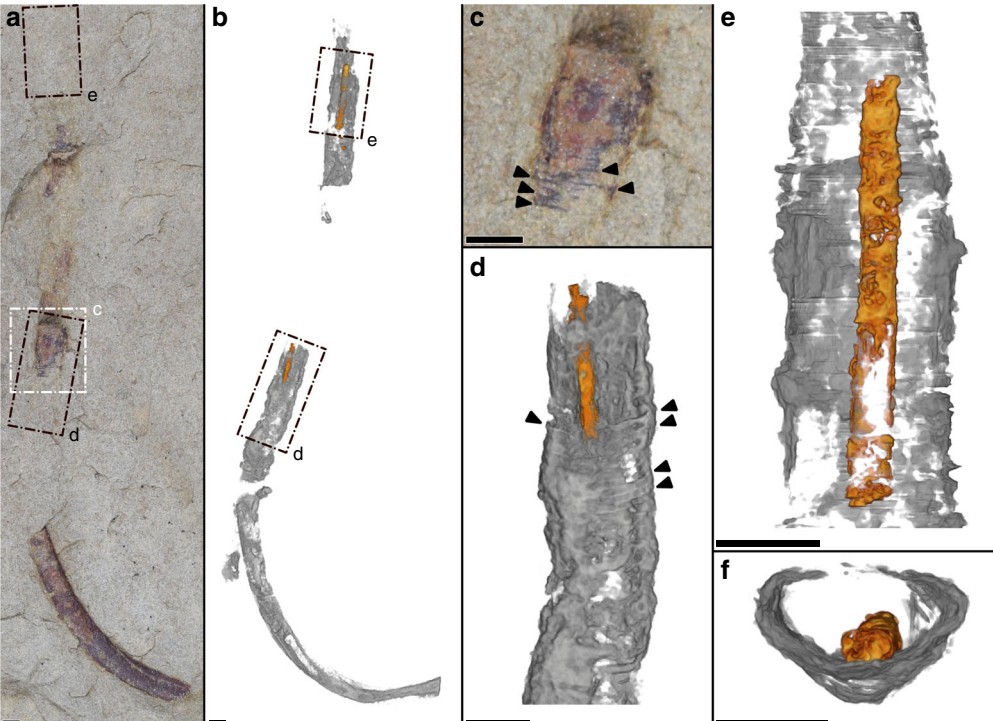

**Fig. 4 Optical imaging and µCT of cloudinomorph pyritized tube and soft tissue. a** Light image of entire specimen (sample USNM-WCF_001) in plan-view, specimen partially obscured at rock surface. **b** Corresponding 3D volume render, showing soft tissue (orange) and tube wall (gray); boxes **d**, **e** are marked in both **a**, **b** to help guide slight differences in orientation. **c** Close-up view of labeled box in **a**, highlighting funnel rims (arrows) on external tube. **d** Close-up view of labeled boxes in **a**, **b**, 3D volume render showing partial soft tissue and funnel rims (arrows); **d** largely overlaps with **c**, but includes also host rock encased portion of the fossil. **e** Partial soft tissue from labeled boxes in **a**, **b**. **f** Cross-sectional view of **e** showing relative position of soft tissue that has settled to the bottom of the external tube wall. Sample reposited at the Smithsonian Institution. All scales = 2 mm.

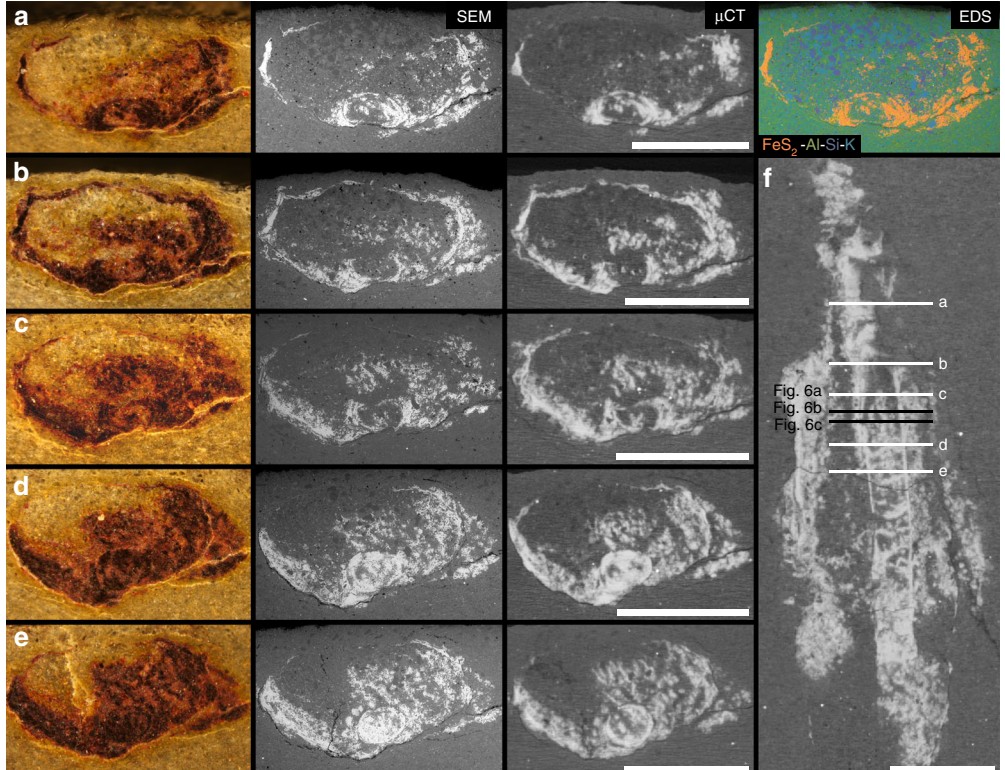

**Fig. 5 Cross-sectional morphology of preserved cloudinomorph soft tissue.** Cross-sections revealed by serial grinding of specimen USNM-WCF_001 illustrated in Fig. 4 (sample USNM-WCF_001); portion of the fossil chosen for grinding shown in Fig. 4e. **a–e** Light and SEM images matched with approximately equivalent µCT tomographic slices (differences in obliquity imposed during serial grinding). Far right in **a** shows tube and gut pyritization via EDS elemental mapping. **f** Position of slices (**a–e** and Fig. 6a–c) shown on µCT tomographic slice through the transverse plane. All scales = 2 mm.

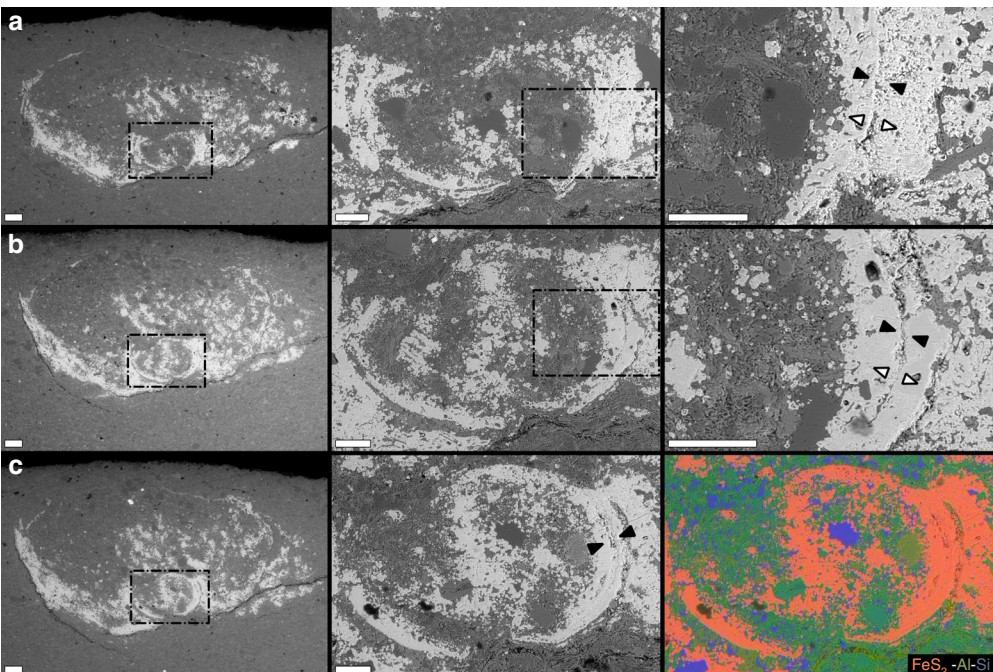

**Fig. 6 Additional detail of cross-sectional morphology.** SEM backscattered electron micrographs (Z-contrast) of specimen USNM-WCF_001, as shown in Figs. 4, 5. Positioning of slices identified in Fig. 5f. Each row corresponds to a single slice at increasing magnifications from left to right (rows **a–c**); dashed boxes in left and middle columns correspond to location for higher magnification images. Right-most frame in row **c** shows EDS elemental map of middle frame in row **c**. Soft tissues in these slices are partially pyrite-infilled (increasingly so from **a** to **c**), though distinct sediment grains can be observed. Note also distinct soft-tissue wall boundaries, indicated by black arrows in higher magnification views. White arrows in higher magnification views of rows **a**, **b** indicate inferred direction of pyrite precipitation from soft-tissue wall, centripetally toward the interior and centrifugally from the exterior. Scales = 200 µm for left-most column, and 100 µm for middle and right-most columns.

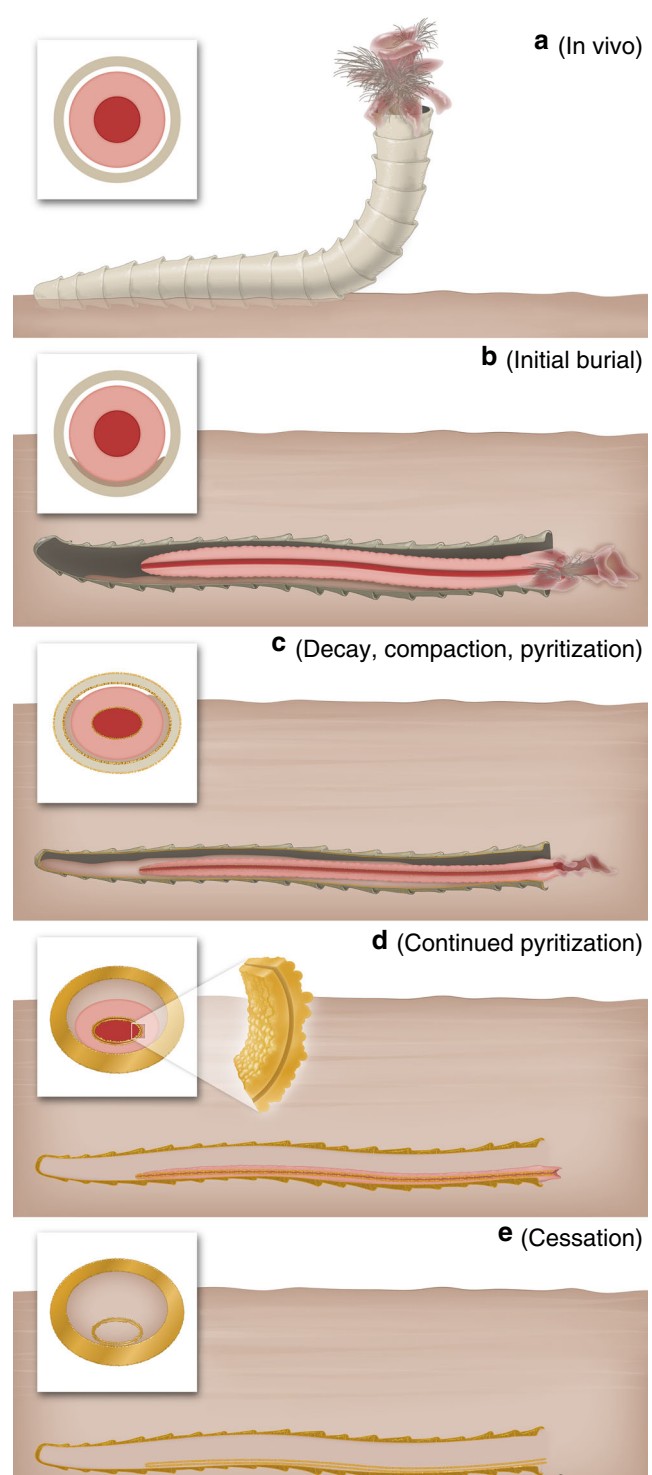

**Fig. 7 Proposed taphonomic sequence of the Wood Canyon cloudinomorph soft tissues. a** Cloudinomorph in hypothesized life position. External soft tissue hypothesized, modeled after siboglinid polychaete. **b** Burial by rapid sedimentation and initiation of decay. Sediment begins to enter tube cavity. **c** Burial compaction of the outer tube from weight of overlying sediment. Early pyritization begins on interior surface of external tube and on both interior and exterior surface of soft-tissue cylinder. **d** Continued pyritization of exterior tube and soft-tissue cylinder. Inset of soft-tissue cylinder wall showing both inward and outward framboidal pyrite growth. **e** Remaining soft tissue decays, leaving pyritized exterior tube and interior soft-tissue cylinder. Gravitational settling of pyritized internal cylinder adjacent to lower external tube boundary. Illustration by Stacy Turpin Cheavens.

concentration was high in the burial setting, pyritization would have therefore been focused more towards the decaying soft tissues[38], resulting in the observed preservational pattern. The kinked soft tissue observed in Fig. 3c may present a slightly different scenario, wherein the organism had died and slumped within its external tube prior to burial positioning or repositioning. And (v), either early sulfate exhaustion caused microbial decay by sulfate reducers to cease, or reduced iron was expended in the burial environment—thus halting continued pyritization. The former chemical limitation may be more realistic. That is, if local sulfate concentrations instead remained sufficient to fuel continued (and less rate-restricted) bacterial sulfate reduction, it is probable that all of the soft tissues of the tube-dweller, including the internal cylindrical structure, would have been more rapidly exhausted. This taphonomic scenario likely would have yielded preservation of the exterior tube with more substantive detail, but leaving no soft tissues to be preserved. We suggest that this is likely the norm for the majority of the specimens recovered from the Wood Canyon Formation (Fig. 2).

**Resolving phylogeny from soft tissue evidence.** In order to provide an improved phylogenetic resolution on the cloudinomorphs, we must first consider which soft tissues are most likely to fossilize. Although they may be rare, there is no shortage of preserved internal soft-tissue structures reported from the fossil record. Fossilized internal soft tissues in the Ediacaran are limited to one possible occurrence of a muscular cnidarian[39]; on the other hand, Cambrian examples are much more numerous and diverse, including cardiovasculature[40], nervous and neurological tissues[41], musculature[42], and copious reports of digestive tracts[43]. In Cambrian lagerstätten, guts are the most frequently preserved internal structures[44]. Whereas fossil vasculature or nervous tissues are preserved as compressed or flattened features[40,41] and musculature as bundled fibrous structures[42], fossil guts can reveal a broadly tubular nature where three-dimensionally preserved, and sometimes occur with the presence of associated digestive glands[43,44]. Cambrian guts are typically preserved either as carbonaceous films[45], sediment infillings[46], or via phosphatization[44], the latter of which is potentially reflective of the organism's digestive physiology. However, there are limited (and perhaps contentious) examples of gut pyritization[47] (but see also ref. [48]) as well as gut-content pyritization[46]. The consistent geopetal nature of the pyritized soft-tissue structures observed here supports the notion that they were originally centrally located structures in vivo, rather than adjacent to the exterior tube wall. At this stage, we can only speculate on the potential histological underpinnings that resulted in preferential pyritization of these features. It is instead their cylindrical expression, propensity for preservation in Cambrian fossils[44,45], and consistent size, shape,

integrity of supporting soft tissues was compromised through decay, the pyritizing soft tissues gravitationally settled to the imposed bottom of the external tube[37]. Thus, both the ventral positioning of the internal cylinders within the recumbent external tubes and the distinction between bedding-respective dorsal and ventral coherency of exterior tube pyritization (or perhaps ventral-inward pyrite infilling) serve as geopetal indicators. The gravitational slumping of the decaying soft tissue within the tube, as oriented recumbently, would have increased the distance for diffusion of bisulfide toward the upward-positioned wall of the exterior tube. If the reduced iron

and position within the external tube that most endorse a gut interpretation (Fig. 7).

Despite being soft tissues, the tendency for gut tracts to be preserved is likely amplified by several factors. Not only can portions of the digestive tract in some organisms be lined with decay-resistant cuticle[43], but guts are also segregated environments hosting their own microbiome and ions sourced from microbial metabolisms and ingested contents at the time of death[44,49]. Guts can thus be isolated and accentuated taphonomic vessels, providing ideal conditions for self-contained mineralization. As observed here, the presence of centripetally precipitated pyrite inward from an apparent soft-tissue cylinder wall suggests that their preservation did indeed proceed from the interior (Fig. 6). The next key challenge is to identify, within reasonable cloudinomorph assignments (Supplementary Table 1) and from both morphological and taphonomic perspectives, which soft tissue structures—whether guts or otherwise—could conceivably leave comparably preserved cylindrical structures (Fig. 8, Supplementary Fig. 1). Below, we detail the two primary but debated assignments for the cloudinomorphs—cnidarians and annelids—and offer supplemental treatment on other possibilities (hemichordates and phoronids, see Supplementary Discussion).

**Cnidarians**. Cnidarians, and more specifically anthozoans, have probably received the most attention as a logical affinity for the cloudinomorphs. Similarities reported between morphological characters of anthozoans and *Cloudina*[15] (Supplementary Table 1) have served to propagate a cnidarian interpretation through the literature. On the other hand, anthozoan internal anatomy is markedly disparate from cylindrical structures observed here. Cnidarians, regardless of class affiliation, are defined in part by the possession of a sac-like gastrovascular coelenteron (Fig. 8a); this simple two-way digestive system has a single orifice for the intake of food and expulsion of waste. Within the anthozoans, the upper portion (the pharynx) can be broadly tubular, opening into a larger, mesentery-lined, and grossly tubular gastrovascular cavity with numerous outpocketings defined by septa, unlike anything observed herein. These numerous septa, which can be calcitic and thus easily preservable, provide structural support of the tubular pharynx and gastrovascular cavity, but such structures are not observed in any cloudinomorphs.

Another possibility that should be considered is that our preserved soft tissues could represent the entire soft-tissue body, rather than an internal feature, of tube-dwelling hydrozoan polyps. Although generally rare and somewhat contentious in the fossil record, hydroid fossils have been reported dating back to the Cambrian[50]. Many hydrozoans live in colonial habits joined by an interconnected network of canals and exterior skeletal branches, for instance, perhaps akin to such modern calcareous examples as *Millepora* fire corals[51]. The cloudinomorph tube construction is strikingly different from the densely porous tubes of the fire corals, but a more important distinction may be found in the pattern of tube branching. If a colonial hydrozoan assignment were fitting for the broader cloudinomorphs, one may expect branching to be more common than observed. Although single-tube branching is known in *Cloudina* and presumed to indicate asexual budding behavior[10,14], it has not been observed in most other comparable tubiform cloudinomorphs, such as those reported here from Nevada[6] and elsewhere[26,52,53]. At last, no indications of tentacles are found in the soft tissues reported herein, which have been considered diagnostic characters in a rigorous evaluation of putative fossil hydrozoans[50]. Although this may pose concern for such an interpretation here, rapid taphonomic loss of tentacles has been shown to be likely[54]. Nevertheless, granting that features of cloudinomorph external

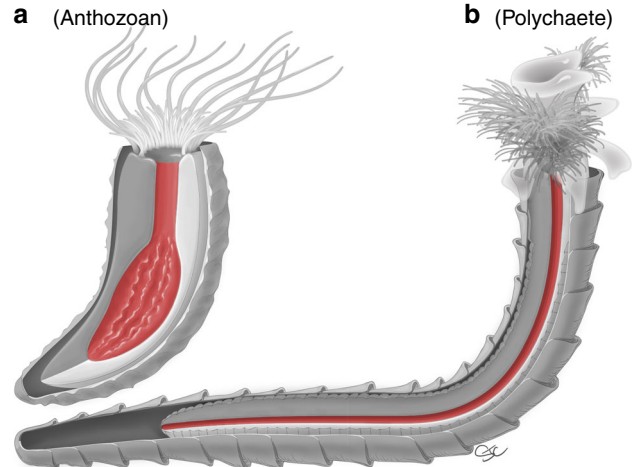

**Fig. 8 Diagrammatic comparison of candidate taxa for cloudinomorph affiliation.** Sections of the tubes and body walls are removed to illustrate gut tracts (red). **a** Anthozoan coelenteron showing upper, tubular pharynx and lower, sac-like gastrovascular cavity with mesentery structure. **b** Polychaete annelid with straight through-gut path. Illustration by Stacy Turpin Cheavens; see also diagrams in Supplementary Fig. 1.

tubes have been deduced to be very generally cnidarian as compared to other plausible affinities[15], the straight, sagittally continuous soft tissues, whether guts or not, are difficult to reconcile in favor of such an affinity.

**Annelids**. The combination of straight, cylindrical soft tissues, and external tube structures may designate polychaete annelid worms as the most fitting phylogenetic position for the cloudinomorphs. Not only do annelid through-guts express simple cylindrical morphologies (Fig. 8b), but the external tubes of the tube-building annelids are also at least structurally comparable to the cloudinomorphs, contrary to previous assertions[15]. For instance, one of the features that has been used as a primary argument against a polychaete affinity[15] is the presence of closed posterior tube ends. Closed ends are known from some posteriorly complete cloudinomorphs, notably *Cloudina*[14] and *Conotubus*[26]; although other cloudinomorphs, like *Saarina*, may have had only partially closed or constricted posterior tube ends[27]. This feature may therefore not be ubiquitous within the cloudinomorphs without clear evidence for a closed basal tube end across all members. Perhaps more importantly, the previous claim[15] that closed bases are absent in modern tube-dwelling polychaetes is unsupported by zoological literature. For example, siboglinids are known to have closed bases[55] and many other tube-dwelling polychaetes possess dedicated anatomical structures (ciliated fecal grooves) or other behavioral strategies to keep waste from accumulating in a closed posterior end of the tube. A second unsubstantiated argument[15] is that polychaete tubes are not composed of nested funnels, but such a tube construction is in fact found in siboglinids like *Oasisia* (Supplementary Fig. 2). Finally, the mode of asexual reproduction by budding as inferred from branching in *Cloudina* tubes[10,14] is sometimes thought to be more indicative of a cnidarian affinity. Tube-dwelling serpulids among other polychaetes, however, are known to undergo comparable clonal reproduction[55]—though not all cloudinomorphs, including those reported here[6], show evidence of external tube branching. The point here is not to invalidate a valuable character evaluation of *Cloudina*[15], but instead to offer caution to its applicability to the broader cloudinomorphs and limited

comparisons with modern tube-dwelling polychaetes. Although the contribution by Vinn and Zatoń[15] effectually compares morphological characters of *Cloudina* to broad-stroke cnidarians, their comparison with tube-dwelling polychaetes, instead, much more narrowly focuses on three sessile, tube-dwelling families—sabellids, serpulids, and cirratulids. The choice of these families clearly results from their calcareous tube-building habits in relation to the tubes of *Cloudina*, but information provided by the fossil record seems incompatible with such comparisons[56]. The records of sabellids and serpulids extend only into the Carboniferous and Triassic[57], respectively, and the cirratulids have a much younger appearance in the Oligocene[58], thus casting doubt on the appropriateness of these families as acceptable comparators.

The overarching phylogenetic systematics of the ecologically diverse annelids is complicated and controversial[59]. They can be generally divided by life mode and feeding strategies into two reciprocal monophyletic major clades—the Errantia (free moving, predatory forms) and the Sedentaria (sessile, tube-dwelling forms)[59]—but they additionally include five basally branching lineages (Oweniidae, Magelonidae, Chaetopteridae, Amphinomidae, and Sipuncula; see Supplementary Fig. 3). The lowest branching of these are tube-dwellers, the oweniids and magelonids[59]. Together, these two families form a monophyletic sister group to the other annelids, the Palaeoannelida[59], followed by the basally branching, tube-building chaetopterids[60].

Outside of the three previously targeted sedentarian families[15], placing the cloudinomorphs within any other specific polychaete designation may still impose a chronological gap, albeit likely more reconcilable, between the terminal Ediacaran and the earliest fossil record of readily identifiable polychaete tubes. The earliest potential examples of polychaete tubes previously reported are indeed Cambrian in age, including organophosphatic chaetopterid tubes (*Hyolithellus*) from Greenland[61] and calcareous tubes of *Coleoides* and *Ladatheca* from Newfoundland and England[62,63]. Although it is important to note that a record of polychaete tubes is ostensibly absent from exceptional Cambrian lagerstätten, such deposits do provide several plausible tube-free annelid fossils, such as (among others) stem-annelids from the Sirius Passet[64]; sipunculids, remarkably similar to recent examples, with preserved gut tracts from the Maotianshan Shale[65]; and numerous polychaetes from the Burgess Shale, most of which preserve gut tracts[45]. Furthermore, moderate taphonomic survival of annelid gut tracts has been demonstrated by decay experiments with polychaetes[37]. These fossils ultimately suggest the divergence of at least the basal-most annelid branches (the palaeoannelids and chaetopterids) within the Cambrian Period[60]. We thus advocate an expanded investigation of the diversity of unresolved but comparable tubiform fossils across the Ediacaran–Cambrian transition[13] in an effort to help potentially connect these records.

**Behavioral considerations.** The structure and ingested contents of fossil guts hold significant potential to be behaviorally and ecologically informative. For instance, the preservation of digestive glandular structure and recognizable prey items in the gut contents of Cambrian ecdysozoans have been used as verification of a predatory or scavenging life mode[43,44,66,67]. These simple cylindrical cloudinomorph soft tissues, however, are lacking any detail of differentiation or compartmentalization—which is not necessarily problematic for a polychaete interpretation[68]. Portions of the soft-tissue cylinder that are fully mineralized, as well as other sections that show sediment infill, can both be resolved with a gut interpretation. First, regions of pyrite infilling of the cloudinomorph guts may tentatively represent mineralization of ingested, non-descript, organic detritus, similar to gut-content/cololite pyritization observed in Cambrian trilobites[46]. Alternatively, these internal gut structures (Figs. 5c, d, 6) may represent pyritized internal gut folds like typhlosoles, which are known to occur in annelids, though the taphonomic resolution and three-dimensional continuity of these features is unfortunately poor. Second, if the observed simple morphology is biologically faithful, in conjunction with their posited sessile habit, then we may be able to deduce that the cloudinomorphs were likely detritivorous and presumably deposit-to-suspension feeders[68]. The flexibility in feeding behaviors of modern-day tube-dwelling polychaetes may provide insight on the presence of sediment encased within these fossil soft tissues. Specifically, *Owenia* and several spionids are among species that can switch between suspension feeding and deposit feeding behaviors depending on external conditions[69]. These organisms are normally suspension feeders in higher current flow, taking food from the water column with their tentacular palps. However, when water current is low and suspended food is unavailable, they tend to employ surface deposit feeding by placing their palps on the surface of the substrate, during which sediment is commonly ingested[69]. This is not meant to suggest that other tube-dwellers could not have behaved similarly, but it is actualistic evidence provided directly by potential modern analogs. The potential feeding flexibility of the cloudinomorphs adds diversity in Ediacaran feeding modes, for example, building on recent suggestions of macroscopic suspension feeding by *Ernietta*[70] and scavenging by motile bilaterians[71].

## Discussion

To our knowledge, the structures reported herein are not only the first recognizable soft tissues in cloudinomorphs, but also the oldest guts yet described in the fossil record. As such, the Wood Canyon tubular fossil assemblage has provided a unique view into early animal anatomy. Nonetheless, for at least the cautions listed throughout the discussion above, we choose to refrain from shoehorning the cloudinomorphs into any explicit polychaete family. However, it is the sum of their parts—including the external tube structure, internal soft tissues, and presumed behavioral considerations—that may best denote placement amongst the Annelida as the most plausible. The accord of sequencing-based phylogenies[59,60] and the available fossil record indicates that stem-annelids, regardless of whether they exhibited a tube-dwelling habit or not, had diverged by at least the early Cambrian—and thus a placement of the terminal Ediacaran cloudinomorphs within basal branches of the annelids is very likely not unreasonable. If these structures are indeed guts, they are the earliest in the record, fortify a terminal Ediacaran presence of bilaterians, demarcate the divergence of the Lophotrochozoa, and, perhaps, help to build a phylogenetic bridge across the Ediacaran–Cambrian boundary to the diversity of annelids known from post-Cambrian Explosion lagerstätten. Nevertheless, when taken together, the novelties provided by the cloudinomorphs in the terminal Ediacaran—including the advent of macroscopic biomineralization[2], the establishment of plausible ecosystem engineering behaviors[8], the enhancement of larval dispersal mechanisms and sexual and asexual reproductive habits[10], plausibly novel feeding strategies, and direct soft-tissue evidence of a through-gut—signpost an immense ecological leap towards the rapid metazoan diversification that transpired geologically soon after.

## Methods

**Sample collection.** The fossils reported here were collected as part of broader studies on the taxonomy[6] and biostratigraphic utility of the tubular fossil assemblages from south-central Nevada[30]. In the Montgomery Mountains, the informal

lower member of the Wood Canyon Formation is predominantly interbedded siltstone and sandstone, interpreted to have been deposited in a shallow marine paleoenvironment[30]. Three shallowing-upward parasequences in the lower member, each capped by dolostone marker beds, provide a regionally consistent stratigraphic framework (Fig. 1). The second dolostone marker contains the nadir of the basal Cambrian negative $\delta^{13}C$ excursion, and immediately underlies the beds bearing the Cambrian GSSP (Global Boundary Stratotype Section and Point) ichnofossil *Treptichnus pedum*[30,72]. The cloudinomorphs were recovered from the first of the parasequences, within a ~5 m siltstone to shale interval below the first dolostone marker[30]. No radiometric ages for these strata currently exist, but a fossil assemblage that includes erniettomorphs, cloudinomorphs, and possible *Swartpuntia* have all been described from the lower member of the Wood Canyon Formation[30,73], together comprising a typical late Ediacaran Nama-type (~550–539 Ma) assemblage.

**X-ray tomographic microscopy**. Primary μCT data collection was conducted using a Zeiss Xradia 510 Versa x-ray microscope. Optimal source conditions, filters, and scan durations varied by sample. All scans were conducted at 80 kV source voltage, 7 W source power, and using the 0.4 × detector objective. Resulting serial x-ray attenuation slices were viewed and 3D reconstructions were conducted with Avizo 9.7 software (Thermo Fisher Scientific) in order to verify coherency, shape, and position of the soft-tissue structures.

**Serial grinding**. Subsequently, one of the soft-tissue-bearing samples was selected for destructive preparation in order to directly view the soft-tissue structure via optical and scanning electron microscopy. We used manual serial grinding (using a Buehler EcoMet250, with grinding intervals of ~0.25 mm) with reflected light microscopy (using a Nikon SMZ1500 binocular microscope with an attached Nikon D600 digital SLR camera) to view the cross-sections of the preserved soft tissues.

**Scanning electron microscopy**. For compositional characterization, we analyzed individual cross-sections using a Zeiss Sigma 500 variable pressure scanning electron microscope (VP-SEM). Backscattered electron images were collected using a high-definition five-segment backscattered electron detector under identical operating conditions (8.5 mm working distance, low vacuum mode (40 Pa chamber pressure, 99.999% nitrogen gas atmosphere), 20 keV beam accelerating voltage, high current mode (40 nA), and a 60 μm aperture). In addition, energy dispersive x-ray spectroscopy (EDS) was conducted using dual, co-planar Bruker 6│30 EDS units integrated on the Sigma 500 VP-SEM, using the same operating conditions as above, with the exception of aperture size (120 μm) to improve x-ray count rate. Count rates were on the order of ~300,000 counts per second, combined from both EDS detectors.

**Reporting summary**. Further information on research design is available in the Nature Research Reporting Summary linked to this article.

## Data availability

All fossil materials will be reposited in the Smithsonian Institution. Sample IDs of fossils with soft-tissue preservation, ordered by appearance in figures: USNM-N1601_FL_018; USNM-E1630_006; USNM-N1601_FL_017; USNM-WCF_001. Data sets generated during the current study, including raw tiff stacks from μCT analyses and images and data files from SEM and EDS analyses, are available from the corresponding authors on reasonable request.

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

## Acknowledgements

We acknowledge helpful discussions with E.P. Anderson and J.W. Huntley, and data processing assistance from B.D. Andreasen. J.D.S. is supported by the National Science Foundation EAR CAREER-1652351; J.D.S. and T.S. are supported by EAR/IF-1636643; S.M.J. is supported by the University of Missouri Preparing Future Faculty Postdoctoral Fellowship; E.F.S. is supported by the National Science Foundation EAR SGP-1827669 and was supported by the Smithsonian Institution Buck Fellowship, the Paleontological Society Research Grant-PA-RG201703, and the NAI Lewis and Clark Fund for Exploration and Field Research in Astrobiology; and L.L.N. was supported by a National Geographic Young Explorer Grant. Illustrations by Stacy Turpin Cheavens, MS, copyright 2019 by The Curators of the University of Missouri, a public corporation. Type and paratype cloudinomorphs from the Wood Canyon Formation reproduced with permission in Fig. 2 from Selly, T. et al. (in press) A new cloudinid fossil assemblage from the terminal Ediacaran of Nevada, USA. *Journal of Systematic Palaeontology*, https://doi.org/10.1080/14772019.2019.1623333, copyright The Natural History Museum, reprinted by permission of Informa UK Limited, trading as Taylor & Francis Group, www.tandfonline.com on behalf of The Natural History Museum.

## Author contributions

Following the CRediT model, J.D.S. is responsible for conceptualization, interpretation, supervision/project administration, SEM analyses, and primary funding acquisition to support the project, with field-collection funding support from E.F.S.; T.S. contributed to conceptualization, performed μCT method development and investigation, and contributed to SEM preparation and investigation with J.D.S.; S.M.J. performed visualization with μCT data and prepared figures with T.S. and J.D.S.; R.A.M. provided expertize on modern invertebrate phylogeny, behavior, and anatomy, and conducted SEM on modern *Oasisia* tubes; E.F.S., L.L.N. and M.A.S. conducted primary field work and sample collection with assistance from J.D.S., T.S. and S.M.J.; Y.C. provided expertize on cloudinomorphs. J.D.S. wrote the manuscript and supplement, with significant input from all of the authors.

## Competing interests

The authors declare no competing interests.
