## [Peer Review File · Nature Communications]

Reviewers' Comments:

Reviewer #1:

Remarks to the Author:

This paper presents new material of terminal Ediacaran cloudinomorpha (putative metazoans living in organic, or more rarely calcified, tubes) from Nevada, which preserve what appears to be the remains of pyritized digestive tracts. The authors describe these structures from several individuals, and then employ both microCT and serial-grinding-based techniques to establish the structure and geometry of the putative guts. The paper concludes that the structures are in all likelihood one-way through guts, lending support to a bilaterian affinity and (broadly) consistent with a placement within the polychaete worms.

This is an exciting paper for a number of reasons – first, the material is remarkable (certainly the first example of preserved Ediacaran internal soft-tissues that I'm aware of), and second, the affinities of cloudinomorpha has been the topic of intense discussion for decades, and these fossils provide a vital piece of the puzzle. The discovery of these internal structures should give evolutionary biologists a lot of hope – now that we know that internal anatomy in these organisms can be preserved, I suspect a lot more people will now be looking, and we may shortly know a lot more about this critical (and unusual) interval in Earth History. Although the structures are almost certainly guts (...there's almost nothing else they could be...) the material and associated interpretations are bound to be the subject of further debate; however, the authors have done a thorough job with the analyses they've employed, and the discussions are carefully reasoned. Consequently, this paper should certainly be published in Nature Communications, and will generate a lot of excitement amongst evolutionary biologists and Precambrian geologists alike. I only have a few minor/moderate comments that may help the authors refine the manuscript; these are numbered below in a way that hopefully makes them easy to address:

1. "stolon-like reproductive modes of other Ediacarans" (line 71); this is a semantic point, but Ediacaran workers have begun to distinguish what have been historically referred to as the 'Ediacara biota' (soft-bodied macroscopic organisms with...probably...a wide variety of affinities), and more plausible metazoans, like Cloudina and other biomineralizing taxa. I think, therefore, that a better way to phrase this might be: "...versus the stolon-like reproductive modes of enigmatic soft-bodied 'Ediacara biota'" or something similar.
2. "Although its cause is equivocal at present, the changing of the ecological guard from largely sedentary Ediacara-type communities to much more dynamic syn-'Cambrian Explosion' ecosystems was well underway in the terminal Ediacaran" (line 74); this is an important point – it might be worth mentioning that Darroch et al. (2018) made the argument that, if all the ecological and evolutionary changes that take place over the Ediacaran and Cambrian are considered together, the biggest step change in organismal and ecological complexity arrives in the latest Ediacaran, rather than at the E-C boundary itself...and thus may represent the first phase of the Cambrian Explosion. Papers by Droser and Tarhan make a similar argument, but place this threshold even earlier, in the White Sea.
3. "The initial burial event reoriented the originally upright cloudinomorpha..." (line 215); this is obviously peripheral to the argument the authors are making, but how sure are we that these organisms were originally upright? I've seen a number of reconstructions for various cloudinomorpha taxa, some which suggest an upright lifestyle...and others which suggest a more recumbent lifestyle. If the inferred feeding behavior (deposit/suspension feeding, depending on conditions) relies on an upright reconstruction, then perhaps this statement requires more justification. Otherwise, the authors could safely acknowledge the uncertainty in this reconstruction.

4. "Detailed comparative analysis of the morphological characters of the external tubes, specifically of Cloudina, has undoubtedly bolstered this case, propagating a cnidarian interpretation through the literature" (line 268); can the authors be more specific here (however briefly) as to what characters on the tubes support a cnidarian affinity? This is key information that readers will appreciate having available as they balance the new interpretations levied in this paper.

5. "...no indications of tentacles are found may pose at least tentative concern for a hydrozoan polyp interpretation—although taphonomic loss could be responsible" (line 289); this is true – Gibson et al. (2018) performed a detailed series of decay experiments on anthozoan cnidarians, and showed that, in Ediacaran-type substrate conditions, tentacles are typically among the first characters lost.

6. "Annelids" (line 352); the observation that some cloudinomorphs branch hints at asexual reproduction. I realize that the authors specifically state that none of their new material from Nevada shows this behavior, but it may be worth mentioning somewhere that some polychaete groups are also able to reproduce asexually. I see that this character is given in the helpful summary table, but I think it bears stating in the text.

7. "If we cannot use the sabellids, serpulids, or cirratulids, however, where can we go (if anywhere) from here?" (line 378); this sentence and the rhetorical question are a bit wordy...and may detract from the important discussion that follows underneath. I would re-write (or remove) this sentence.

8. "we may be able to deduce that the cloudinomorphs were likely detritivorous and presumably deposit-to-suspension-feeders" (line 415); I think this is an important point, and speaks to the larger ecological significance of the gut structures. I think the authors are maybe missing a small trick by not expanding briefly on what this might mean for the evolution and diversity of Precambrian feeding modes – recent papers by Gibson et al. (2019) (who demonstrate macroscopic suspension feeding by *Ernietta*, also in the latest Ediacaran), and Gehling and Droser (2018) (who suggest bilaterian scavenging as early as ~550 Ma) are demonstrating that Ediacaran ecosystems may have been far more complex (and Cambrian-like) than originally thought. The presence of abundant metazoan suspension feeders in shallow marine environments throughout the late Ediacaran hints at ecological complexity and patterns of nutrient cycling that are a million miles away from what we thought as little as 10 years ago. I'm not encouraging the authors to arm-wave, but I think a brief couple of sentences to this effect could add to the impact of the paper.

9. Along the same lines as point 8 above, I found myself wondering about the hypothesized excretory system of these organisms. I know that non-tube dwelling polychaetes can be prodigious producers of fecal pellets (which aren't, admittedly, a common or well-established feature of the Ediacaran trace fossil record), but I don't know how tube-dwelling forms manage to excrete. If they do produce fecal pellets (or some other type of waste that might conceivably stand a chance of being preserved in the sedimentary record – perhaps remineralized as peloids?) it would be an interesting prediction to make, and might widen the search window for supporting evidence.

10. Figure 5; I found this figure fascinating...and I'm wondering if there are isn't additional structure within the preserved guts that may be biologically meaningful. For example, in panel rows c) through d) the interpreted gut lumen seems to be split in two, with (perhaps?) preserved soft tissue collected in the middle. I'm wondering whether this might be the remains of typhlosole tissue?

Reviewer #2:

Remarks to the Author:

Dear authors and editor,

I have concluded my review of the manuscript entitled "Tube-dwelling animals of the terminal Ediacaran Period reveal the oldest fossil guts" submitted by James D. Schiffbauer and others for publication in Nature Communications. This interesting article present for the first time a real good evidence of soft tissue preservation in an organism from the Ediacaran Nama Assemblage, shedding new light on the evolution and phylogeny of cloudinomorpha.

All investigative methods utilized (μ CT, optical and scanning electron) clearly reveal a cylindrical structure adjacent to the internal surface of the tube, that are putatively interpreted as preserved guts. In my point of view, the inner structure does not represent a taphonomic artifact and can be easily attributed to a preserved soft-tissue due to: 1) proven occurrence in several samples; 2) distinct preservation from those observed in the exterior tube; 3) all the inner structures analyzed present similar shape and dimension; and 4) all the inner cylindrical tubes seems to follows the full length of the external tubes. Following this evidences, the most plausible hypothesis is that the inner tube is preserved guts or even the full (soft) body of the organism.

I have been studying examples of cloudinomorpha from South America, China and Namibia, and the (few) evidences I have of soft tissue preservation indicates that the organism filled the entire tube. In the examples presented, the inner tube always occurs in the lower portion following the outer tube wall, which leads me to believe that it corresponds only to a part of the original organism. Therefore, I think the hypothesis that these objects are actually preserved guts is the most parsimonious. The taphonomic and preservational model presented are consistent and help to explain the distinct preservation between soft tissue and the outer part of the tube.

Considering the presence of preserved guts in the specimens analyzed all the discussion of cloudinomorpha phylogeny seems to point to an already suggested annelid affinity. However, some key points need to be explored further as: 1) evidences of asexual reproduction in Cloudina (presence of daughter tubes with equal diameter) it is a characteristic much more associated with cnidarian than annelid (Hua et al. 2005); and 2) the granular microstructure in Cloudina shell also points to a cnidarian grade (Vinn & Zatón 2012). I think that a deepening in these topics could reinforce the hypothesis of cloudinomorpha affinity with annelids.

Finally, based on the considerations above, I consider article innovative and acceptable for publication in Nature communications after minor revision. However, it is fundamental to critically review and re-evaluate the affinity of cloudinomorpha with annelids considering the points above mentioned. Some minor questions and suggestions are pointed in the PDF archive attached.

I sincerely hope that these few comments are useful for the improvement of this manuscript and I also congratulate the authors on this interesting piece of science.

I look forward to seeing the revised version in press.

Best wishes,

References

HUA H., CHEN Z., YUAN X., ZHANG L., XIAO S. 2005. Skeletogenesis and asexual reproduction in the earliest biomineralizing animal Cloudina. *Geology*, 33:277-280.

VINN O. & ZATÓN M. 2012. Inconsistencies in proposed annelid affinities of early biomineralized. Organism Cloudina (Ediacaran): structural and ontogenetic evidences. *Carnets de Geologie*: 39-47 (CG2012_A03).

Reviewer #3:

Remarks to the Author:

Recommendation: Accepted for publication after minor revision.

Please see attached file for detailed review.

Rudy Lerosey-Aubril.

REVIEW OF Schiffbauer *et al.*

Title: Tube-dwelling animals of the terminal Ediacaran Period reveal the oldest fossil guts

Authors: J.D. Schiffbauer, T. Selly, S.M. Jacquet, R.A. Merz, L.L. Nelson, M.A. Strange, Y. Cai, and E.F. Smith.

Reviewer: R. Lerosey-Aubril (Harvard University)

Recommendation: Accepted after minor revision.

General Comment

The contribution of Schiffbauer *et al.* documents the preservation of internal structures within terminal Ediacaran tubiform fossils from Nevada, USA. The authors argue that these structures represent three-dimensional remains of a simple tubular digestive tract, which have been preserved via post-mortem bacterially-induced precipitation of pyrite (iron disulfide). If correct, this interpretation would indicate that 1) the evolution of a through gut dates back to the terminal Ediacaran, and 2) the fossils from Nevada – and similar forms known as Cloudinomorpha – were amongst the oldest bilaterians (animals exhibiting a bilateral symmetry).

The terminal Ediacaran is a critical period of the history of Life. It separates a time when the seafloor was inhabited by (mostly) large enigmatic organisms (the “Ediacaran biotas”) from the rapid diversification of early bilaterians and the emergence of modern-style marine ecosystems (the “Cambrian Explosion”). It is also associated with the evolution of the first biomineralized skeletons and complex behavioral strategies (as evidenced by the ichnofossil

record). In this context, demonstrating that some terminal Ediacaran fossils were vermiform bilaterans would be of the utmost scientific significance.

Some aspects of the fossils and their preservation are a bit problematic to me, but I believe that the authors present a strong case for interpreting the internal structures present in these tubes as the remains of straight digestive tracts. Although not completely unexpected with regard to the ichnofossil record, this is an important result which contributes to a better understanding of what might have been the initial stage of one of the most important events of the history of life – the rise of animals. Accordingly, **I strongly recommend the publication of this article in Nature Communications after minor revision.**

Data, Methodology, and Conclusions

The authors used a great variety of techniques (SEM, EDS, μ CT) to investigate these important fossils and the internal structures they may or may not (89% of the specimens!) preserve. This allows a particularly detailed description of these otherwise relatively simple structures and the way they are preserved. This profusion of details may sometimes impact a little bit the flow of arguments presented by the authors, but I personally found that it really helps building a convincing case in favor of their interpretation as digestive structures. I do not think of any equipment or methodology that the authors could have used to strengthen their view. I also greatly appreciated that they do not overinterpret their data and fully acknowledge when some of their observations were difficult to explain or weakened some of their arguments. **The conclusions are sound and as robust as can be when investigating c. 540 Myr-old internal organs of soft-bodied organisms.**

Text

Overall, I am happy with the organization of the manuscript, but I am not as convinced as the authors that identifying the affinities within the Bilateria of the Nevadan fossils is that important. The description of the fossils and the discussions on the preservation and ecology are interesting and appropriate in length, but **I believe the section on the phylogenetic affinities could be significantly reduced.**

In the main text, I would restrict the discussion to why they believe annelid affinities is the most likely hypothesis (detailed discussion) and briefly summarize why other affinities are less likely; a reference to a supplementary text discussing more extensively the alternative hypotheses (=the subsections ‘Anthozoans’, ‘Hemichordates’, and ‘Phoronids’) would be enough to me. This would permit a better balance between the three discussions (preservation/affinities/ecology).

Illustrations

The illustrations are of good quality and really help understanding the text. However, the article could possibly benefit from:

- Adding a simple interpretative drawing next to the 3D volume-render illustrating the internal soft tissue for each specimen on Figure 4.
- Making sure that Figure 4 is published at a sufficiently large size (full-page width or close to). The labels on this figure seem much larger than on the other figures, which suggests its original size is much smaller – I think that if intentional, this is a mistake.

- **Adding a figure composed of a series of simple diagrams illustrating the various steps of the preservational model.**

More specific comments

The aim of the following comments/questions is to help the authors improving some parts of their text. They do not need to be blindly followed, just considered with serious.

Line 135: I have noticed that the specimens with the guts in Figs. 3 and 4 are all notably larger than the ones illustrated in Fig. 2 (the only ones clearly identifiable as cloudinomorphs). I would recommend that you add the size (diameter) range of the Nevadan tubes somewhere in this first paragraph to show that the ones preserving internal structures are not special in this regard (or are they?).

Line 137: Please spell Lagerstätte(n) with a capital letter (German word).

Lines 154-156: I would be more cautious here, for I only see evidence of annulation in Fig. 3. Maybe you could rephrase along those lines: “We interpret the annulation of the tubes observed both optically and by microCT as a vestige of a ‘funnel-in-funnel’ organization, which supports the hypothesis of their cloudinomorphs affinities”.

Lines 176-177: any idea why this difference between bottom and upper edges?

Lines 181-184: “centripetally precipitated pyrite”. Do you have any idea what property of the gut wall could explain that its internal and external surfaces are the sites of nucleation of pyrite?

The only times I observed pyrite crystals within fossilized guts (e.g. Zhu et al. 2014 – your ref. 44), they were frambroids scattered within the gut content.

Lines 181-184: I think you should measure the thickness of the ‘discernable cylinder wall’ and check whether it is compatible with the known thickness of a gut wall in... annelids for instance.

Line 251: I would be more nuanced here, for I do not think that it is correct to say that “the majority of Cambrian guts are phosphatized”. The majority of gut structures in Cambrian Lagerstätten (Burgess Shale Type mostly) are preserved as carbon films. Sediment-like infillings are also particularly common in some of these sites (Chengjiang). Gut phosphatisation is indeed frequently observed in many of these sites, but not all (e.g. Chengjiang – in part due to taphonomic history), and only in arthropods (likely due to physiological reasons). Even in arthropod fossils with phosphatized gut structures, these are frequently associated with carbonaceous remains of other parts of the digestive system.

Lines 256-258: I would strongly recommend to cite the remarkable paper of Butler et al. here:

Butler, A. D., Cunningham, J. A., Budd, G. E., & Donoghue, P. C. (2015). Experimental taphonomy of *Artemia* reveals the role of endogenous microbes in mediating decay and fossilization. *Proceedings of the Royal Society B: Biological Sciences*, 282(1808), 20150476.

Line 407: My apologies for the self-promotion, but I think the following paper of mine would be worth citing here:

Zacai, A., Vannier, J., & Lerosey-Aubril, R. (2016). Reconstructing the diet of a 505-million-year-old arthropod: *Sidneyia inexpectans* from the Burgess Shale fauna. *Arthropod Structure & Development*, 45(2), 200-220.

Fig. 5: Do you have any idea why the tube wall is absent or very thin where the internal tube/gut abuts it? Is it just local or similar observations can be made in other parts of this specimen or other specimens?

Rudy Lerosey-Aubril – Cambridge, 13/9/19.

RESPONSE TO REVIEWERS

The following text has been compiled from the reviewers' remarks to the authors. We have provided a response to each comment as appropriate to indicate our adjustments or reasoning for not adjusting the text in question. Our responses are in **BLUE/BOLD** font.

Reviewer #1 (Remarks to the Author):

This paper presents new material of terminal Ediacaran cloudinomorpha (putative metazoans living in organic, or more rarely calcified, tubes) from Nevada, which preserve what appears to be the remains of pyritized digestive tracts. The authors describe these structures from several individuals, and then employ both microCT and serial-grinding-based techniques to establish the structure and geometry of the putative guts. The paper concludes that the structures are in all likelihood one-way through guts, lending support to a bilaterian affinity and (broadly) consistent with a placement within the polychaete worms.

This is an exciting paper for a number of reasons – first, the material is remarkable (certainly the first example of preserved Ediacaran internal soft-tissues that I'm aware of), and second, the affinities of cloudinomorpha has been the topic of intense discussion for decades, and these fossils provide a vital piece of the puzzle. The discovery of these internal structures should give evolutionary biologists a lot of hope – now that we know that internal anatomy in these organisms can be preserved, I suspect a lot more people will now be looking, and we may shortly know a lot more about this critical (and unusual) interval in Earth History. Although the structure are almost certainly guts (...there's almost nothing else they could be...) the material and associated interpretations are bound to be the subject of further debate; however, the authors have done a thorough job with the mix analyses they've employed, and the discussions are carefully reasoned. Consequently, this paper should certainly be published in Nature Communications, and will generate a lot of excitement amongst evolutionary biologists and Precambrian geologists alike.

RESPONSE: We thank the reviewer for their excitement on our findings and their detailed comments below. We have done our best to incorporate all suggestions where appropriate.

I only have a few minor/moderate comments that may help the authors refine the manuscript; these are numbered below in a way that hopefully makes them easy to address:

1. "stolon-like reproductive modes of other Ediacarans" (line 71); this is a semantic point, but Ediacaran workers have begun to distinguish what have been historically referred to as the 'Ediacara biota' (soft-bodied macroscopic organisms with...probably...a wide variety of affinities), and more plausible metazoans, like Cloudina and other biomineralizing taxa. I think, therefore, that a better way to phrase this might be: "...versus the stolon-like reproductive modes of enigmatic soft-bodied 'Ediacara biota'" or something similar.

RESPONSE: adjusted as suggested.

2. "Although its cause is equivocal at present, the changing of the ecological guard from largely sedentary Ediacara-type communities to much more dynamic syn-'Cambrian Explosion' ecosystems was well underway in the terminal Ediacaran" (line 74); this is an important point – it might be worth mentioning that Darroch et al. (2018) made the argument that, if all the ecological and evolutionary changes that take place over the Ediacaran and Cambrian are considered together, the biggest step change in organismal and ecological complexity arrives in the latest Ediacaran, rather than at the E–C boundary itself...and thus may represent the first phase of the Cambrian Explosion. Papers by Droser and Tarhan make a similar argument, but place this threshold even earlier, in the White Sea.

RESPONSE: We agree with the importance of this statement, and have added one brief line to

this effect here – as follows: “Although its cause is equivocal at present, the changing of the ecological guard from largely sedentary Ediacara-type communities to much more dynamic syn-‘Cambrian Explosion’ ecosystems was well underway in the terminal Ediacaran. Indeed, as recently proposed⁴, this interval possibly displays an even larger step-change in organismal and ecological complexity than at the Ediacaran-Cambrian boundary itself.”

3. “The initial burial event reoriented the originally upright cloudinomorpha...” (line 215); this is obviously peripheral to the argument the authors are making, but how sure are we that these organisms were originally upright? I’ve seen a number of reconstructions for various cloudinomorpha taxa, some which suggest an upright lifestyle...and others which suggest a more recumbent lifestyle. If the inferred feeding behavior (deposit/suspension feeding, depending on conditions) relies on an upright reconstruction, then perhaps this statement requires more justification. Otherwise, the authors could safely acknowledge the uncertainty in this reconstruction.

RESPONSE: Agreed. This passage has been rephrased to indicate uncertainty in this reconstruction – as follows: “The initial burial event emplaced the cloudinomorpha within the sulfate-reduction zone of the sediment (oriented prone to bedding, whether³⁴ or not³⁵ this was their in vivo position).”

4. “Detailed comparative analysis of the morphological characters of the external tubes, specifically of *Cloudina*, has undoubtedly bolstered this case, propagating a cnidarian interpretation through the literature” (line 268); can the authors be more specific here (however briefly) as to what characters on the tubes support a cnidarian affinity? This is key information that readers will appreciate having available as they balance the new interpretations levied in this paper.

RESPONSE: We do agree with this statement, but we had already included both a list of several features in the introduction, as well as a supplementary table. At this point in the text, we decided it better to refer the reader back to the table as opposed to providing what we feel would be a redundant list of features.

5. “...no indications of tentacles are found may pose at least tentative concern for a hydrozoan polyp interpretation—although taphonomic loss could be responsible” (line 289); this is true – Gibson et al. (2018) performed a detailed series of decay experiments on anthozoan cnidarians, and showed that, in Ediacaran-type substrate conditions, tentacles are typically among the first characters lost.

RESPONSE: Agreed, we have edited the text accordingly and added the appropriate reference – as follows: “Lastly, no indications of tentacles are found in the soft tissues reported herein, which have been considered diagnostic characters in a rigorous evaluation of putative fossil hydrozoans⁵⁰. While this may pose concern for such an interpretation here, rapid taphonomic loss of tentacles has been shown to be likely⁵⁴.”

6. “Annelids” (line 352); the observation that some cloudinomorpha branch hints at asexual reproduction. I realize that the authors specifically state that none of their new material from Nevada shows this behavior, but it may be worth mentioning somewhere that some polychaete groups are also able to reproduce asexually. I see that this character is given in the helpful summary table, but I think it bears stating in the text.

RESPONSE: Now stated more directly – as follows: “Finally, the mode of asexual reproduction by budding as inferred from branching in *Cloudina* tubes^{10,14} is sometimes thought to be more indicative of a cnidarian affinity. Tube-dwelling serpulids among other polychaetes, however, are indeed known to undergo comparable clonal reproduction⁵⁵—though not all cloudinomorpha, including those reported here⁶, show evidence of external tube branching. The point here is not to invalidate a valuable character evaluation of *Cloudina*¹⁵, but instead to offer caution to its applicability to the broader cloudinomorpha and limited comparisons with modern tube-dwelling polychaetes.”

7. “If we cannot use the sabellids, serpulids, or cirratulids, however, where can we go (if anywhere) from here?” (line 378); this sentence and the rhetorical question are a bit wordy...and may detract from the important discussion that follows underneath. I would re-write (or remove) this sentence.

RESPONSE: Agreed and removed.

8. “we may be able to deduce that the cloudinomorpha were likely detritivorous and presumably deposit-to-suspension-feeders” (line 415); I think this is an important point, and speaks to the larger ecological significance of the gut structures. I think the authors are maybe missing a small trick by not expanding briefly on what this might mean for the evolution and diversity of Precambrian feeding modes – recent papers by Gibson et al. (2019) (who demonstrate macroscopic suspension feeding by *Ernietta*, also in the latest Ediacaran), and Gehling and Droser (2018) (who suggest bilaterian scavenging as early as ~550 Ma) are demonstrating that Ediacaran ecosystems may have been far more complex (and Cambrian-like) than originally thought. The presence of abundant metazoan suspension feeders in shallow marine environments throughout the late Ediacaran hints at ecological complexity and patterns of nutrient cycling that are a million miles away from what we thought as little as 10 years ago. I’m not encouraging the authors to arm-wave, but I think a brief couple of sentences to this effect could add to the impact of the paper.

RESPONSE: We have added a closing sentence to this section to illustrate the importance of increasing feeding complexity – as follows: “The potential feeding flexibility of the cloudinomorpha adds diversity in Ediacaran feeding modes, for example, building on recent suggestions of macroscopic suspension feeding by *Ernietta*⁷⁰ and scavenging by motile bilaterians⁷¹”

9. Along the same lines as point 8 above, I found myself wondering about the hypothesized excretory system of these organisms. I know that non-tube dwelling polychaetes can be prodigious producers of fecal pellets (which aren’t, admittedly, a common or well-established feature of the Ediacaran trace fossil record), but I don’t know how tube-dwelling forms manage to excrete. If they do produce fecal pellets (or some other type of waste that might conceivably stand a chance of being preserved in the sedimentary record – perhaps remineralized as peloids?) it would be an interesting prediction to make, and might widen the search window for supporting evidence.

RESPONSE: This is certainly an interesting thought-experiment, but unfortunately one that is based (at present) on a lack of evidence. As such, we don’t feel that this paper is the right location to expound on these possibilities.

10. Figure 5; I found this figure fascinating...and I’m wondering if there are isn’t additional structure within the preserved guts that may be biologically meaningful. For example, in panel rows c) through d) the interpreted gut lumen seems to be split in two, with (perhaps?) preserved soft tissue collected in the middle. I’m wondering whether this might be the remains of typhlosole tissue?

RESPONSE: We agree, and have added a line of text to note this possibility – as follows: “Alternatively, these internal gut ‘structures’ (Figs. 5c–d and 6) may represent pyritized internal gut folds like typhlosoles, which are known to occur in annelids, though the taphonomic resolution and three-dimensional continuity of these features is unfortunately poor.”

Reviewer #2 (Remarks to the Author):

Dear authors and editor,

I have concluded my review of the manuscript entitled “Tube-dwelling animals of the terminal Ediacaran Period reveal the oldest fossil guts” submitted by James D. Schiffbauer and others for publication in Nature Communications. This interesting article present for the first time a real good evidence of soft tissue preservation in an organism from the Ediacaran Nama Assemblage, shedding new light on the evolution and phylogeny of cloudinomorpha.

All investigative methods utilized (μ CT, optical and scanning electron) clearly reveal a cylindrical structure adjacent to the internal surface of the tube, that are putatively interpreted as preserved guts. In my point of view, the inner structure does not represent a taphonomic artifact and can be easily attributed to a preserved soft-tissue due to: 1) proven occurrence in several samples; 2) distinct preservation from those observed in the exterior tube; 3) all the inner structures analyzed present similar shape and dimension; and 4) all the inner cylindrical tubes seems to follows the full length of the external tubes. Following this evidences, the most plausible hypothesis is that the inner tube is preserved guts or even the full (soft) body of the organism.

I have been studying examples of cloudinomorphs from South America, China and Namibia, and the (few) evidences I have of soft tissue preservation indicates that the organism filled the entire tube. In the examples presented, the inner tube always occurs in the lower portion following the outer tube wall, which leads me to believe that it corresponds only to a part of the original organism. Therefore, I think the hypothesis that these objects are actually preserved guts is the most parsimonious. The taphonomic and preservational model presented are consistent and help to explain the distinct preservation between soft tissue and the outer part of the tube.

Considering the presence of preserved guts in the specimens analyzed all the discussion of cloudinomorphs phylogeny seems to point to an already suggested annelid affinity. However, some key points need to be explored further as: 1) evidences of asexual reproduction in *Cloudina* (presence of daughter tubes with equal diameter) it is a characteristic much more associated with cnidarian than annelid (Hua et al. 2005); and 2) the granular microstructure in *Cloudina* shell also points to a cnidarian grade (Vinn & Zatón 2012). I think that a deepening in these topics could reinforce the hypothesis of cloudinomorphs affinity with annelids.

Finally, based on the considerations above, I consider article innovative and acceptable for publication in *Nature communications* after minor revision. However, it is fundamental to critically review and re-evaluate the affinity of cloudinomorphs with annelids considering the points above mentioned. Some minor questions and suggestions are pointed in the PDF archive attached.

I sincerely hope that these few comments are useful for the improvement of this manuscript and I also congratulate the authors on this interesting piece of science. I look forward to seeing the revised version in press.

RESPONSE: We thank the reviewer for their positive evaluation and comments here, as well as in the appended pdf. First, we have corrected the locality/stratigraphy figure as suggested in the pdf. Second, with regard to the exploration of points 1 and 2 above, the rearrangement of the manuscript per the request of reviewer #3 allowed us to spend a bit more text on one of these issues. Specifically, we did rework the passage on asexual reproduction, which appears in the cnidarian subsection. While we discussed the granular microstructure within the introduction, we've decided not to lengthen this discussion further simply because two of the authors here have another accepted manuscript at *Scientific Reports* evaluating the microstructure of other cloudinomorphs. It seems to us that we would effectively be stealing our own thunder should we spend a significant amount of text on this argument – which isn't central to whether our structures are preserved soft tissues.

Reviewer #3 (Remarks to the Author):

Recommendation: Accepted for publication after minor revision.
Please see attached file for detailed review.

Rudy Lerosey-Aubril.

[COPIED FROM ATTACHED PDF]

REVIEW OF Schiffbauer *et al.*

Title: Tube-dwelling animals of the terminal Ediacaran Period reveal the oldest fossil guts

Authors: J.D. Schiffbauer, T. Selly, S.M. Jacquet, R.A. Merz, L.L. Nelson, M.A. Strange, Y. Cai, and E.F. Smith.

Journal: *Nature Communications*

Reviewer: R. Lerosey-Aubril (Harvard University)

Recommendation: Accepted after minor revision.

General Comment

The contribution of Schiffbauer *et al.* documents the preservation of internal structures within terminal Ediacaran tubiform fossils from Nevada, USA. The authors argue that these structures represent three-dimensional remains of a simple tubular digestive tract, which have been

preserved via post-mortem bacterially-induced precipitation of pyrite (iron disulfide). If correct, this interpretation would indicate that 1) the evolution of a through gut dates back to the terminal Ediacaran, and 2) the fossils from Nevada – and similar forms known as Cloudinomorpha – were amongst the oldest bilaterians (animals exhibiting a bilateral symmetry).

The terminal Ediacaran is a critical period of the history of Life. It separates a time when the seafloor was inhabited by (mostly) large enigmatic organisms (the “Ediacaran biotas”) from the rapid diversification of early bilaterians and the emergence of modern-style marine ecosystems (the “Cambrian Explosion”). It is also associated with the evolution of the first biomineralized skeletons and complex behavioral strategies (as evidenced by the ichnofossil record). In this context, demonstrating that some terminal Ediacaran fossils were vermiform bilaterians would be of the utmost scientific significance.

Some aspects of the fossils and their preservation are a bit problematic to me, but I believe that the authors present a strong case for interpreting the internal structures present in these tubes as the remains of straight digestive tracts. Although not completely unexpected with regard to the ichnofossil record, this is an important result which contributes to a better understanding of what might have been the initial stage of one of the most important events of the history of life – the rise of animals. Accordingly, ***I strongly recommend the publication of this article in Nature Communications after minor revision.***

RESPONSE: We thank Dr. Lerosey-Aubril for his evaluation and positive remarks, and have responded accordingly to his queries below (where appropriate).

Data, Methodology, and Conclusions

The authors used a great variety of techniques (SEM, EDS, μ CT) to investigate these important fossils and the internal structures they may or may not (89% of the specimens!) preserve. This allows a particularly detailed description of these otherwise relatively simple structures and the way they are preserved. This profusion of details may sometimes impact a little bit the flow of arguments presented by the authors, but I personally found that it really helps building a convincing case in favor of their interpretation as digestive structures. I do not think of any equipment or methodology that the authors could have used to strengthen their view. I also greatly appreciated that they do not overinterpret their data and fully acknowledge when some of their observations were difficult to explain or weakened some of their arguments. **The conclusions are sound and as robust as can be when investigating c. 540 Myr-old internal organs of soft-bodied organisms.**

Text

Overall, I am happy with the organization of the manuscript, but I am not as convinced as the authors that identifying the affinities within the Bilateria of the Nevadan fossils is that important. The description of the fossils and the discussions on the preservation and ecology are interesting and appropriate in length, but **I believe the section on the phylogenetic affinities could be significantly reduced.**

In the main text, I would restrict the discussion to why they believe annelid affinities is the most likely hypothesis (detailed discussion) and briefly summarize why other affinities are less likely; a reference to a supplementary text discussing more extensively the alternative hypotheses (=the subsections ‘Anthozoans’, ‘Hemichordates’, and ‘Phoronids’) would be enough to me. This would permit a better balance between the three discussions (preservation/affinities/ecology).

RESPONSE: We agree, and have moved discussion of hemichordates and phoronids to a supplemental discussion document. Contrary to Dr. Lerosey-Aubril’s suggestion here, however, we do leave discussion of the Cnidarians/Anthozoans in the main text, as this is one of the primary phylogenetic placements considered for the cloudinomorpha in previous literature. It seems most appropriate to have these compared in conjunction with the annelids in the body of the article.

Illustrations

The illustrations are of good quality and really help understanding the text. However, the article could possibly benefit from:

- Adding a simple interpretative drawing next to the 3D volume-render illustrating the internal soft tissue for each specimen on Figure 4.

RESPONSE: Agreed, and done. We've replaced the right frame of each panel from the original with interpretive diagrams. The contrasting filters used previously did not add much to the visibility of the soft-tissue structures.

- Making sure that Figure 4 is published at a sufficiently large size (full-page width or close to). The labels on this figure seem much larger than on the other figures, which suggests its original size is much smaller – I think that if intentional, this is a mistake.

RESPONSE: We agree, and have made this a full page width figure.

- Adding a figure composed of a series of simple diagrams illustrating the various steps of the preservational model.

RESPONSE: Agreed. We commissioned our scientific illustrator to prepare this figure for us, and agree that it is a significant addition to the manuscript.

More specific comments

The aim of the following comments/questions is to help the authors improving some parts of their text. They do not need to be blindly followed, just considered with serious.

Line 135: I have noticed that the specimens with the guts in Figs. 3 and 4 are all notably larger than the ones illustrated in Fig. 2 (the only ones clearly identifiable as cloudinomorpha). I would recommend that you add the size (diameter) range of the Nevadan tubes somewhere in this first paragraph to show that the ones preserving internal structures are not special in this regard (or are they?).

RESPONSE: This is an excellent suggestion. We have added some information from our forthcoming *Journal of Systematic Palaeontology* manuscript to address the size range, and have noted that those with guts tend to be on the larger end of the range (at least above median, but not outside of the observed size range).

Line 137: Please spell Lagerstätte(n) with a capital letter (German word).

RESPONSE: We have capitalized Lagerstätte where referring to a specific deposit (i.e., Gaojiashan Lagerstätte), but have not done so where generally referring to exceptional deposits.

Lines 154-156: I would be more cautious here, for I only see evidence of annulation in Fig. 3. Maybe you could rephrase along those lines: “We interpret the annulation of the tubes observed both optically and by microCT as a vestige of a ‘funnel-in-funnel’ organization, which supports the hypothesis of their cloudinomorpha affinities”.

RESPONSE: Adjusted nearly as suggested (replaced ‘organization’ here with ‘tube construction’).

Lines 176-177: any idea why this difference between bottom and upper edges?

RESPONSE: This question has now been addressed in the Taphonomic Model section. The gravitational slumping of the decaying soft tissue within the tube, as oriented recumbently, would have increased the distance for diffusion of bisulfide towards the upward (with respect to bedding) wall of the exterior tube, hence making the lower portion more prone to pyritization if in a high-iron setting.

Lines 181-184: “centripetally precipitated pyrite”. Do you have any idea what property of the gut wall could explain that its internal and external surfaces are the sites of nucleation of pyrite? The only times I observed pyrite crystals within fossilized guts (e.g. Zhu et al. 2014 – your ref. 44), they were frambroids scattered within the gut content.

RESPONSE: This was briefly mentioned in the Taphonomic Model section, such that the gut wall was histologically suitable for pyritization. We'd prefer not to speculate further as to why the gut would be more suitable.

Lines 181-184: I think you should measure the thickness of the 'discernable cylinder wall' and check whether it is compatible with the known thickness of a gut wall in... annelids for instance.

RESPONSE: While we agree that this is a great idea, it is unfortunately muddled by presumed pyrite replacement and decay of the original gut wall – both of which would result in distortion of the original thickness. Effectively, what makes this feature most discernable is an apparent line (in cross-section) between pyrite with evident growth outwardly and inwardly. If viewing just this line, it provides the cylindrical boundary of the gut.

Line 251: I would be more nuanced here, for I do not think that it is correct to say that “the majority of Cambrian guts are phosphatized”. The majority of gut structures in Cambrian Lagerstätten (Burgess Shale Type mostly) are preserved as carbon films. Sediment-like infillings are also particularly common in some of these sites (Chengjiang). Gut phosphatisation is indeed frequently observed in many of these sites, but not all (e.g. Chengjiang – in part due to taphonomic history), and only in arthropods (likely due to physiological reasons). Even in arthropod fossils with phosphatized gut structures, these are frequently associated with carbonaceous remains of other parts of the digestive system.

RESPONSE: We have amended this part in the discussion to better reflect available data from the fossil record.

Lines 256-258: I would strongly recommend to cite the remarkable paper of Butler et al. here: Butler, A. D., Cunningham, J. A., Budd, G. E., & Donoghue, P. C. (2015). Experimental taphonomy of *Artemia* reveals the role of endogenous microbes in mediating decay and fossilization. *Proceedings of the Royal Society B: Biological Sciences*, 282(1808), 20150476.

RESPONSE: We agree, and have added this citation.

Line 407: My apologies for the self-promotion, but I think the following paper of mine would be worth citing here:

Zacai, A., Vannier, J., & Lerosey-Aubril, R. (2016). Reconstructing the diet of a 505-million-year-old arthropod: *Sidneyia inexpectans* from the Burgess Shale fauna. *Arthropod Structure & Development*, 45(2), 200-220.

RESPONSE: No problem at all – we agree, and have added this citation.

Fig. 5: Do you have any idea why the tube wall is absent or very thin where the internal tube/gut abuts it? Is it just local or similar observations can be made in other parts of this specimen or other specimens?

RESPONSE: This appears to be only a local phenomenon where we were able to section the fossil. In μ CT virtual sections, there are several other regions with a robust lower tube-wall bound. We have added a brief statement to the text so that others that notice the same trend in the figure are aware.

Reviewers' Comments:

Reviewer #1:

Remarks to the Author:

The authors have done an excellent job with the revision, and in my opinion the manuscript is ready to be published. - Simon Darroch

Reviewer #2:

Remarks to the Author:

Dear authors and editor,

I consider that all the suggestions made by me have been fully answered in this version of the manuscript. Therefore, I consider this version suitable for publication in Nature. I look forward to the published version of this important piece of science.

Best regards,

Lucas Warren

Reviewer #3:

Remarks to the Author:

Dear authors,

Thanks for carefully considering my comments and suggestions of changes. Your manuscript looks great, congratulations to you and your illustrator (nice new figure!). I really look forward to reading the published version.

Kind regards,

Rudy Lerosey-Aubril

RESPONSE TO REVIEWERS' COMMENTS:

Reviewer #1 (Remarks to the Author):

The authors have done an excellent job with the revision, and in my opinion the manuscript is ready to be published. - Simon Darroch

Reviewer #2 (Remarks to the Author):

Dear authors and editor,

I consider that all the suggestions made by me have been fully answered in this version of the manuscript. Therefore, I consider this version suitable for publication in Nature. I look forward to the published version of this important piece of science.

Best regards,

Lucas Warren

Reviewer #3 (Remarks to the Author):

Dear authors,

Thanks for carefully considering my comments and suggestions of changes. Your manuscript looks great, congratulations to you and your illustrator (nice new figure!). I really look forward to reading the published version.

Kind regards,

Rudy Lerosey-Aubril

RESPONSE: We would like to offer a general statement of our gratitude to all of the reviewers for their time, expert suggestions and opinions, and meticulousness. This manuscript has benefited greatly from your efforts.